# A Comparative Study on Microstructure and Properties of Ultra-High-Speed Laser Cladding and Traditional Laser Cladding of Inconel625 Coatings

**DOI:** 10.3390/ma15186400

**Published:** 2022-09-15

**Authors:** Yuhang Ding, Wenya Bi, Cheng Zhong, Tao Wu, Wanyuan Gui

**Affiliations:** National Center for Materials Service Safety, University of Science and Technology Beijing, Beijing 100083, China

**Keywords:** laser cladding, coatings, Inconel625, wear resistance, finite element method

## Abstract

In this study, ultra-high-speed laser cladding (UHSLC) and traditional low-speed laser cladding (LSLC) were employed to prepare high-quality Inconel625 coatings on 27SiMn substrates. UHSLC has cladding speeds of 30 m/min, which are 15 times faster than those of LSLC, and it produces a much greater cladding efficiency, which is 13.9 times greater than LSLC. The microstructure of the Inconel625 coatings was investigated in detail utilizing field emission scanning electron microscopy (FESEM) and electron probe microanalyzer (EPMA). According to the FESEM results, UHSLC Inconel625 coatings have more refined crystals than LSLC Inconel625 coatings. Nevertheless, the EPMA results indicate that the UHSLC Inconel625 coatings exhibit much more severe elemental segregation. Moreover, the hardness, wear and corrosion resistance of Inconel625 coatings are significantly enhanced by increasing the laser cladding speed. Furthermore, the reasons for the differences in microstructure and properties of Inconel625 coatings prepared by UHSLC and LSLC were clarified by finite element simulation. UHSLC technique is, therefore, more suitable for preparing Inconel625 coatings on 27SiMn steel surfaces than LSLC.

## 1. Introduction

Industrial parts are constantly in demand, and researchers are increasingly interested in parts designed to withstand harsh conditions. A hydraulic support system made of 27SiMn is an indispensable part of underground mining [1]. The hydraulic support must possess high wear resistance, hardness, and corrosion resistance in order to withstand the complex service environment in the well. Up to date, hard chrome coatings have been applied to hydraulic support components for wear and corrosion protection using traditional electroplating methods [2,3,4]. Hard chrome coatings have been increasingly restricted due to the presence of hexavalent chrome (Cr^6+^), which is considered carcinogenic and hazardous to the environment by the World Health Organization [5]. Environmentally friendly technologies are highly recommended for replacing chrome-faced hard chrome coatings and improving energy efficiency and product performance.

Laser cladding is the most promising technology for surface modification due to the fact that it provides abundant layers of high-quality, compact, and crack-free coatings along with extraordinary physical and chemical properties [6,7,8]. Unfortunately, traditional laser cladding technique is limited by the powder feeding mode and low laser cladding speeds. The physical and chemical properties of coatings fabricated by traditional laser cladding techniques degrade significantly since large amounts of heat introduced into the substrate will cause the coating to incorporate multiple elements in the substrate. In addition, most of the laser energy is absorbed by the substrate because of the low cladding speed, which results in irreparable damage to thin components [9]. An innovative technology (UHSLC) was proposed by Fraunhofer ILT in 2016, which offers numerous advantages in terms of cladding efficiency and coating strength. A variable parameter can be adjusted to achieve cladding speeds between 20 m/min and 200 m/min, allowing coating thicknesses to range between 10 and 250 μm. High-speed laser cladding technology can provide high yields and high efficiency when compared to lower-speed laser cladding [10,11,12,13]. By redesigning the coaxial powder feeding nozzle, powder particles could be heated to melting point before being guided into the molten pool. The low laser-transmitted energy enabled a micro-molten pool to be created on the substrate, which allowed the coating to be prepared with a low dilution ratio and metallurgical bonding and the coating to provide an excellent combination of mechanical and chemical characteristics [14,15,16,17].

Due to the high wear resistance and corrosion resistance requirements of hydraulic supports, Fe-based alloys and Ni-based alloys are commonly used for laser cladding on their surface [6,18,19,20,21]. Ouyang et al. [22] conducted a laser cladding experiment using a Fe-based alloy coating on the surface of 27SiMn steel substrate, and the results demonstrated that the corrosion resistance of the alloy coating was higher than that of the substrate, resulting in an improvement of the corrosion resistance of the hydraulic support. Wang et al. [23] investigated the microstructure and properties of Fe-based alloy coating using laser cladding with conventional gravity feeding (GF) and high-speed powder feeding (HF). According to the results, the microstructure of HF coating consists of uniform, small grains. In addition, the hardness of the HF coating is 9.4% higher than that of the GF coating, and the wear amount of the HF coating is reduced by 80.5%. Bai et al. [24] prepared a Fe-based alloy coating on 27SiMn steel pipe, and investigated the phase composition, hardness, and wear mechanism at different regions of the coating. Liu et al. [25] prepared a nickel-based composite coating (Ni60-Ti_3_SiC_2_) on 304 stainless steel. According to the results, the coating had a considerable improvement in hardness and wear resistance due to the fine grain and dispersion strengthening caused by the small-sized hard phases. Gui et al. [26] prepared an Inconel625 coating on Q245R steel by high-speed laser cladding technique and found that the coating was better at resisting corrosion and wear than that of the substrate. Scendo et al. [27] prepared an Inconel625 coatings on S235JR by using wire (WIn/S) and powder (PIn/S), respectively. The results indicated that the coatings prepared with powder had a better hardness and corrosion resistance. In spite of the substantial progress that has been made, most of these works have been conducted on traditional laser cladding technique, and improvements in substrate surface performance can largely be attributed to the application of high-performance coating materials. Yet, the reasons for the differences in microstructure and properties of the same coatings materials prepared by UHSLC and LSLC are not well understood.

Inconel625 has outstanding mechanical properties and excellent corrosion resistance, so creating an Inconel625 coating on the surface of 27SiMn steel can significantly enhance 27SiMn steel’s wear resistance and corrosion resistance [28,29]. UHSLC and LSLC were used in this work to prepare high-quality Inconel 625 coatings on 27SiMn substrates, with cladding speeds of 30 m/min and 2 m/min, respectively. The reason for differences in microstructure and properties of Inconel625 coatings prepared by UHSLC and LSLC were clarified by field emission scanning electron microscopy (FESEM), X-ray diffraction (XRD), electron probe microanalyzer (EPMA) and finite element simulation.

## 2. Experimental Section

### 2.1. Materials Preparation

Commercial-grade Inconel625 powder (Höganäs, Sweden) was used to perform the subsequent UHSLC and LSLC process. As shown in Figure 1, Inconel625 alloy powder is spherical, ensuring the continuity of the powder feeding process during laser cladding. As shown in Figure 1b, the particle size distribution of Inconel625 powder was analyzed using a laser diffraction particle size analyzer (Mastersizer 3000, Malvern, UK). Inconel625 alloy powder used in this work displayed a particle size distribution primarily between 24.1 and 98.1 μm, with an average particle size of 59.3 μm. A 27SiMn steel pipe with an inner diameter of 85 mm and an outer diameter of 95 mm is used as a base material during the laser cladding process. Before cladding, the pipe is polished and cleaned with alcohol. The chemical composition of 27SiMn steel and Inconel625 powder is presented in Table 1.

### 2.2. Laser Cladding

The heat source is provided by a fiber laser system (ZKZM-4000, ZKZM, Xi’an, China), and the laser beam spot diameter is 2 mm. The schematic diagram of the laser cladding experimental platform is shown in Figure 2, which is comprised of three parts: a high-speed lathe, a laser powder feeding system and a 27SiMn steel pipe. The powder feeding system and laser are integrated into the laser head at the front end of the robot, and the preset motion trajectory is set through programming control. During the laser cladding process, the high-speed machine tool drives the pretreated 27SiMn steel pipe to rotate, and the laser head is driven by the robot to move along the steel axis. The Inconel625 powder was fed into a laser molten pool by coaxial feeding driven by argon at a flow rate of 1.4 L/min. The laser melts the powder and a small amount of base material, forming a molten pool on the surface of the steel pipe. The experiment was carried out in a protective chamber where the steam of argon was continuously supplied at a flow rate of 14.5 L/min in order to prevent oxidation of the molten metal. After laser removal, self-cold solidification obtains a uniform coating covering the entire surface of the steel shaft. The optimal processing parameters, such as laser power, powder feeding rate, cladding speed, and overlap rate were carried out as shown in Table 2.

### 2.3. Characterization

After the cladding is complete, the sample was cut into a 10 × 10 × 10 mm size. The section of the sample was first ground to 2000 grit using silicon carbide paper and then polished with diamond paste to a mirror finish. Ultrasonic-assisted was used to thoroughly clean the surface with deionized water and alcohol. All polished cross-sections were etched several times in aqua regia solution (HCl:HNO_3_ = 3:1), and then cleaned with deionized water. An optical metallographic microscope (OM) and a scanning electron microscope (SEM) equipped with an energy spectrometer (EDS) were used to analyze the cladding layer’s microstructure. White light interferometers were used to measure the surface roughness of samples. A scan speed of 5 °/min was used for X-ray diffraction (XRD) to identify all samples, ranging from 20° to 120°. Cu Kα (λ = 1.54 Å) radiation was used in all the patterns, and graphite monochromator was used as a filter. Electron probe microanalyzer (EPMA) was used to determine the element distribution of coatings.

A Vickers hardness tester was used along the vertical bonding line, with a load of 5 N and a continuous measurement of the microhardness distribution. Multiple measurements were made to reduce experimental error. CETR-UMT-2MT was used to conduct a linear friction and wear test of the coating in the length direction. The testing was carried out using reciprocating sliding mode. The coupling parts were Si_3_N_4_ ceramic balls, hardness grade 8.8, load 25 N, single sliding length 10 mm, sliding speed 10 mm/s and sliding time 30 min. Corrosion resistance of Inconel625 coatings was characterized by electrochemical testing using a three-electrode system (electrolyte with Pt sheet as counter electrode and KCl electrode as reference electrode). The specimens were encapsulated in rosin within a 30 mm PVC tube. During encapsulation, only one side of the specimens was exposed to the outside environment. NaCl solution (3.5 wt%) was used as the electrolyte.

## 3. Results and Discussion

### 3.1. Surface and Cross-Sectional Microstructure Feature

The microstructural evolution is described by two main parameters (the growth rate R and the temperature gradient G). As shown in Figure 3, the value of G/R may cause a change in the solidification method, while the value of product G*R determines the size of the solidification structure. Laser scanning of the surface of the substrate will create a molten pool, which will solidify when the laser beam leaves the molten zone. When the G/R ratio decreases, one type of solidified structure will then change from planar grain to cellular grain, then columnar dendrites, and finally equiaxed dendrites [30,31].

Figure 4 compares LSLC and UHSLC in terms of cladding efficiency and heat input. The cladding speed of LSLC is 2 m/min, the cladding efficiency is 13.3 cm^2^/min, and the heat input is 66 KJ/m. By comparison with LSLC, UHSLC has significantly faster cladding speeds, and the cladding speed is 30 m/min. With the speed advantage, the cladding efficiency is greatly improved, with the cladding rate reaching 184.93 cm^2^/min, which is 13.9 times as efficient as LSLC. Furthermore, the energy input is decreased, with the heat input reaching only 4.4 KJ/m, which is 6% as efficient as LSLC, therefore directly affecting the microstructure.

Figure 5 depicts SEM images of Inconel625 alloys coatings deposited by UHSLC and LSLC. The Figure 5a,d illustrate the surface morphology of the Inconel625 alloys coatings prepared by UHSLC and LSLC, respectively. On the surface of the Inconel625 alloys coatings prepared by LSLC sticky powder appears, and even powder agglomeration appears. For the Inconel625 alloys coatings prepared by UHSLC, however, there are very few sticky powders on the Inconel625 alloys coatings surface, mostly single particles. There are two reasons why a significant amount of un-melted powder adheres to the surface of LSLC Inconel625 alloys coatings. On the one hand, due to an increased heat input of per unit area in the LSLC process, when compared to UHSLC Inconel625 alloys coatings surfaces, LSLC Inconel625 alloys coatings surfaces have higher temperatures after cladding. On the other hand, as opposed to UHSLC Inconel625 alloys coatings surface, the LSLC Inconel625 alloys coatings surface has a longer contact time with powder when cladding the same-sized area. Figure 5b,e show the cross-sectional microstructure near the clad/substrate interface at different laser scanning speeds. There is no doubt that both UHSLC and LSLC can form metallurgical bonds with their cladding layers. The high growth rate results in a low-to-moderate G/R ratio and pushes the microstructure towards a cellular structure that appears at the bottom of both UHSLC Inconel625 alloys coatings [30,31]. LSLC Inconel625 alloys coatings have higher solidification rates (R) and more moderate temperature gradients than UHSLC Inconel625 alloys coatings since heat is introduced into the substrate when a lower laser scanning speed is used. Thus, columnar dendrites can be seen in the bottom zone of Inconel625 alloys coatings prepared by LSLC. Figure 5c,f show the cross-sectional microstructure near the top coatings surface at different laser scanning speeds. As a consequence of being exposed to the air, the top zone of UHSLC Inconel625 alloys coatings has a much higher solidification rate, which promotes the formation of equiaxed dendrites. Alternatively, local microsegregation may lower the melting point and can increase nucleation along the dendritic front, inhibiting the growth of columnar crystals [30,31], while the columnar dendrites are observed for LSLC Inconel625 alloys coatings due to the relatively lower temperature gradient caused by the lower laser scanning speed introducing more heat.

Figure 6 depicts the surface roughness of the Inconel625 alloys coatings measured by a white-light interferometer prepared by UHSLC and LSLC. UHSLC Inconel625 alloy coatings show relatively low surface roughness; the roughness of UHSLC Inconel625 alloys coatings is 51.90% of the LSLC Inconel625 alloys coatings. Error bars are caused by systemic deviations in the experiments and software processing. The results are due to the fact that during the UHSLC process, the molten pool was small, the powder melted to a sufficient level, and the cladding coating formed was thin and uniform. A smooth and high-quality surface enables UHSLC Inconel625 alloys coatings to be applied directly to a wide range of industrial applications without extensive grinding and turning [29].

Coating thickness is an important consideration in evaluating a coating. An analysis is presented of the thickness of the coating under various conditions, and the results are illustrated in Figure 7. A coating with good formability was obtained under each of the test parameters. The images indicate that the thickness of Inconel625 alloys coatings prepared by UHSLC is generally less than that of Inconel625 alloys coatings prepared by LSLC. The thickness of the Inconel625 alloys coatings prepared by LSLC is 408.45 μm, while the thickness of the Inconel625 alloys coatings prepared by UHSLC is only 106.67 μm. This phenomenon is likely to be explained by the longer thermal processing time and higher heat input during LSLC process [32,33].

### 3.2. Phase Composition Characterization

Fast heating and rapid cooling are two characteristics of laser cladding, which leads to a non-equilibrium solidification and, thus, leads to a supersaturated solid solution and phase lattice distortions. Traditional laser cladding is different from high-speed laser cladding in both cladding speed and cooling rate, thereby affecting phase composition of cladding coatings [9]. The XRD patterns for the Inconel625 alloys coatings prepared by UHSLC and LSLC are shown in Figure 8. Careful indexations of the pattern of Inconel625 alloys coatings prepared by UHSLC and LSLC indicate that the Inconel625 alloys coatings mainly consist of the γ-Ni phase. Comparing the peak positions of pure Ni to 44.5°, 51.9°, 76.4°, 92.9° and 98.5° [34], both coatings have slightly smaller peak positions. Based on the Bragg equation, the crystal plane spacing of the coating is relatively large, which results from the distortion of the lattice.
*2dsinθ = nλ,*(1)

The results show that differences in laser cladding speed only affect preferred orientation for crystal growth of Inconel625 alloys coatings and the main peak change to (200) from (111) as the form changes to UHSLC from LSLC [35].

A crystal size peak and dislocations can be evaluated using XRD analysis. According to the Debye-Scherrer equation [36,37,38], the average crystallite size was calculated based on the XRD peak width of the peak position of the main orientation:(2)$$D=\frac{K\lambda }{\beta cos\theta },$$
where β is the full width at half maxima (FWMH), *K* is a constant equal to 0.94, λ is the wavelength of the incident X-ray (λ = 1.54 Å), *D* is the crystallite size, and θ is the Bragg angle. A comparison of the results of the average grain size calculation for different levels of remelting power is presented in Table 2.

The dislocation density *δ* is defined as the number of dislocations in a unit volume crystal, which can be calculated using the equation below:(3)$$\delta \;=\frac{1}{D^{2}},$$

According to the following formula, the lattice constant was calculated:(4)$$a\;=d_{hkl}\sqrt{h^{2}+k^{2}+l^{2}},$$

According to the calculated results, UHSLC has a smaller grain size and a higher dislocation density than LSLC. It is determined by the process characteristics in which the cladding rate of UHSLC is accompanied by the rate of solidification, both of which are extremely fast.

### 3.3. Elements Distribution of Coatings Characterization

A uniform distribution of elements is very important for laser cladding coatings to achieve optimum surface performance. The use of UHSLC has been widespread for several years now; however, little attention has been given to its distribution of coating elements. Figure 9 illustrates the distribution of the main elements (Fe, Ni, Cr) in the Inconel625 alloys coatings prepared by UHSLC and LSLC. As the primary element (Ni) in the Inconel625 alloys coatings is different from the principal element (Fe) in the 27SiMn steel pipe substrate, the boundary between the coating and the substrate is clearly defined. UHSLC Inconel625 alloys coatings have obvious element segregation problems. Clearly, the element segregation problem is a common phenomenon to UHSLC, which can be attributed to two factors. Firstly, the speed of the steel pipe is too fast, resulting in a strong centrifugal effect and a very high cooling rate, resulting in a melted coating layer that is not evenly distributed and solidified. Moreover, due to the temperature of the molten pool, the consolidation of the bonding interface is controlled by the viscosity μ, which is linked to the temperature T of the molten pool. The calculation is as follows [39]: (5)$$\mu \;=\frac{16}{15}\left(\sqrt{\frac{m}{k_{b}T}\gamma }\right),$$
where m is the atomic mass, $k_{b}$ is the Boltzmann’s constant, and γ is the liquid surface tension of the steel matrix. Under LSLC, the laser beam brings additional heat input, which increases the temperature of the molten pool and, therefore, reduces μ. Therefore, the fluidity of the molten pool will be improved, which will facilitate the diffusion of elements in the Inconel625 alloys coatings. As a result, the LSLC Inconel625 alloys coatings possess a more uniform element distribution. The calculation of the relevant temperature field will be described in detail in the following sections.

### 3.4. Surface Mechanical Properties

Figure 10 illustrates the distribution of cross-sectional hardness for Inconel625 alloys coatings prepared by UHSLC (Figure 10a) and LSLC (Figure 10b). It has been observed that the maximum microhardness of surfaces of LSLC Inconel625 alloy coatings is 309.4 HV_0.5_, whereas the average microhardness of surfaces of UHSLC coatings is 324.4 HV_0.5_. As a result, the maximum hardness of UHSLC alloy coatings is 4.85% greater than that of LSLC alloy coatings. Upon crossing the fusion line, the hardness values for the Inconel625 alloys coatings are close, and the hardness decreases in a similar manner. Under UHSLC, the hardness of the Inconel625 alloys coatings HAZ is higher than under LSLC because the Inconel625 alloys coating is slightly thicker under LSLC, and, thus, has a larger heat affected zone. It is possible that the higher cooling rate of UHSLC results in finer and more uniform microstructure, increasing the Inconel625 alloys coating’s microhardness [26,32].

The friction and wear tests were conducted on Inconel625 alloys coatings prepared by UHSLC and LSLC. The relationship between friction coefficient and time was compared between UHSLC and LSLC coatings. The change in friction coefficient with time is illustrated in Figure 11. The friction coefficient fluctuates up and down with time, and the same coating has different friction coefficients under different cladding speeds, as shown in Figure 11a,b. In order to better observe the fluctuation of the friction coefficient, a line is drawn at the friction coefficient of 0.6. From the figure, it can be seen that the friction and wear process can be divided into two distinct stages: running-in wear and stable wear. The results show that the friction coefficient for UHSLC Inconel625 alloys coatings fluctuates between 0.5 and 0.8, which is smaller than the friction coefficient (0.6–0.8) for LSLC Inconel625 alloys coatings.

Figure 12 illustrates the wear track morphologies of Inconel625 alloys coatings prepared by UHSLC and LSLC after the test., respectively. UHSLC Inconel625 alloys coatings wear surface has deep, long and uniform grooves, and there is a small amount of wear debris on wear surface, indicating that the main mechanism of sliding wear is cutting wear, and there is a certain amount of wear debris wear as shown in Figure 12a. In Figure 12b, there is considerable wear debris on the wear surface of the LSLC Inconel625 alloys coatings, which indicates that wear debris is the primary wear mechanism at this time and cutting wear is secondary. Accordingly, fewer additional elements were introduced into the UHSLC Inconel625 alloys coatings than into the LSLC Inconel625 alloys coatings. Dynamic recrystallization during high-speed laser cladding (involving dislocations, tangles, and sub-grains) can additionally enhance the wear resistance of Inconel625 alloys coatings [40,41].

The wear mass of the Inconel625 alloys coatings is illustrated in Figure 13. The wear mass for the Inconel625 alloys coatings under UHSLC and LSLC is 2.89 mg and 4 mg, respectively. With the change in the friction coefficient, the UHSLC Inconel625 alloys coatings has greater wear resistance than LSLC Inconel625 alloys coatings. Error bars are caused by systemic deviations in the experiments and instruments. The reason for this is that when the cladding speed increases, the large columnar crystals are transformed into small equiaxed dendrites through grain refinement [42,43]. 

The corrosion resistances of UHSLC and LSLC Inconel625 alloys coatings were tested by electrochemical method based on the three electrodes system. As shown in Figure 14, the polarization curves of Inconel625 alloys coatings under UHSLC and LSLC are given to evaluate its corrosion resistance. For UHSLC Inconel625 coating, the corresponding self-corrosion potential is 0.19 V, and the corresponding self-corrosion current density is 1.99 × 10^−6^ A/cm^2^. For LSLC Inconel625 alloys coating, the corresponding self-corrosion potential is −0.04 V, and the corresponding self-corrosion current density is 2.64 × 10^−6^ A/cm^2^ (Table 3). Self-corrosion potential is only a concept of thermodynamics, and dynamic characteristics of materials must also be considered. Current corrosion should be taken into account when evaluating capacity. The corrosion current of a material is determined by its dissolution rate. In general, the smaller the corrosion current, the more corrosion resistant the cladding layer. In contrast, self-corrosion current density for the UHSLC Inconel625 alloys coatings decreased by 24.6%. This suggests that the UHSLC process can enhance corrosion resistance.

To evaluate the corrosion resistance of the Inconel625 alloys coatings, we calculated their corrosion rates. It is necessary to calculate the alloy equivalent weight (EW) in order to determine the corrosion rate [44,45]. Suppose a unit mass of alloy has been oxidized. The electron equivalent *Q* for 1 g of an alloy is:(6)$$Q\;=\sum \frac{nifi}{Wi},$$
where, *fi* = the mass fraction of the *i*th element in the alloy, *Wi* = the atomic weight of the *i*th element in the alloy and *ni* = the valence of the *i*th element of the alloy. Therefore, the alloy equivalent weight, *EW*, is the reciprocal of this quantity:(7)$$EW\;=\frac{1}{\sum \frac{nifi}{Wi}},$$

At this point, we can use Faraday’s Law to calculate the corrosion rate of the reaction.
(8)$$CR\;=K_{1}\;\frac{i_{corr}}{\rho }EW,$$
where, *K*_1_ = 3.27 × 10^−3^, mm g/μA cm year and  ρ = density in g/cm^3^. In accordance with the above formula, we calculated the corrosion rates of Inconel625 alloys prepared by UHSLC and LSLC, respectively. The corrosion rate for UHSLC Inconel625 alloys coatings is 2.09 × 10^−2^ mm/year, while the corrosion rate for LSLC Inconel625 alloys coatings is 2.78 × 10^−2^ mm/year. The corrosion rate of UHSLC Inconel625 alloys coatings has been reduced by 24.82% compared to that of LSLC Inconel625 alloys coatings (Table 4). These results demonstrate that UHSLC can be effective in improving the corrosion resistance of Inconel625 alloys coatings. Similar results have been reported previously [4,5], indicating that the better corrosion resistance was due to ultra-high-speed laser cladding that produced a finer and more uniform texture.

### 3.5. Numerical Simulation of Temperature Field in Laser Cladding Process

Laser cladding is characterized by rapid movement of the laser, which results in rapid heating and rapid solidification, as well as a number of chemical and physical reactions [46,47]. In addition to requiring substantial resources, the experimental approach does not accurately record temperature changes during cladding. To illustrate this, this paper applies the finite element method to simulate UHSLC and LSLC processes, examines the temperature field changes at different speeds of cladding and provides theoretical support for the actual microstructure analysis.

A Gaussian surface heat source is used in this study as the heat source model, and its function is as follows [45,48,49,50]:(9)$$q\left(r\right)=\frac{3Q}{{\pi r}_{H}^{2}}\exp\left(-\frac{3r^{2}}{r_{H}^{2}}\right),$$
where Q is the density of the heat source, r is the moving distance of the laser beam, r_H_ is the radius of the laser beam. The basis material in this paper is structural steel, the powder is Inconel625 powder, the density is 7850 kg/m^3^, the thermal conductivity is 60.5 W/m∙°C, and the specific heat is 434 J/Kg∙°C.

As the high-speed laser cladding process produces cladding on the tube, we take a very small part of the steel tube in order to observe the cladding process. Considering that the part is of small volume, it can be viewed roughly as a cube. The UHSLC and LSLC patterns are established, including the substrate and the cladding layer, the surface of the substrate is defined as the XY plane, and the laser moving direction is along the +X axis. In general, the substrate is defined as a cube with dimensions of 5 mm in length, 5 mm in width and 2.5 mm in height, with a curved surface section with a length of 5 mm, a height of 0.1 mm and a width of 1 mm for the UHSLC coating. For purposes of LSLC coating, a single pass is defined as a surface cut of 5 mm in height, 4 mm in width and 1 mm in height.

With regard to the cladding process, we define the laser power as 2200 W and its radius as 1 mm. For UHSLC, the cladding speed is defined as 30 m/min; for LSLC, the cladding speed is defined as 2 m/min. During the cladding process, Figure 15 illustrates the temperature distribution of the substrate at various times. In conclusion, the UHSLC brings less heat to the substrate, which is consistent with the previous calculation results. In UHSLC, the maximum temperature of the model is 1542.1 °C, whereas in LSLC, the maximum temperature is 4739.3 °C, which is 307.33% of UHSLC. In addition, it is noted that the T_max_ of UHSLC and LSLC coatings is approximately 3000 °C and 1200 °C, respectively. UHSLC requires 1/15 as much time (0.01 s) as LSLC (0.15 s) to travel the same distance.

In order to obtain a more intuitive understanding of the temperature field changes of laser cladding at different speeds, the trend of temperature changes over time at three points (points 1, 2 and 3) on the same line on the substrate was examined, as shown in Figure 16. Due to its high-speed capabilities, UHSLC has a much lower maximum temperature experienced than for the same spot under the LSLC, as well as a quicker cooling rate. Based on the actual cladding layer height, it can be speculated that under the same output power, as the cladding speed increases, the laser energy being applied to the cladding layer will decrease, which will directly result in a reduction of the actual cladding layer height and the heat-affected zone.

## 4. Conclusions

(1)As compared to LSLC, UHSLC Inconel625 alloy coating exhibits reduced powder sticking, roughness and coating thickness. The smoother surface and more efficient production process make UHSLC technology more suitable for practical applications;(2)A higher speed of cladding leads to a faster cooling rate of the coating. A high laser cladding rate increased the value of R (growth rate of dendritic crystal tips), and a high laser cladding rate increased the value of R (growth rate of dendrite tips), which resulted in a decrease in G/R (microstructure supercooling) and an equiaxed dendrite structure in the microstructure;(3)By increasing the laser cladding speed, the cladding structure is continuously refined, and coarse columnar crystals are transformed into fine dendrites. Hardness of the coating has been greatly improved, as well as wear resistance and corrosion resistance.(4)Despite the improvement in UHSLC coating performance, the melted coating elements are unevenly distributed and solidified due to the strong centrifugal effect and very high cooling rate, which should be addressed in the future;(5)A comparison of the heat input and cladding efficiency of UHSLC and LSLC was conducted. Simulations based on finite elements were conducted for the two, allowing the distributions of their temperature fields to be determined.

## Figures and Tables

**Figure 1 materials-15-06400-f001:**
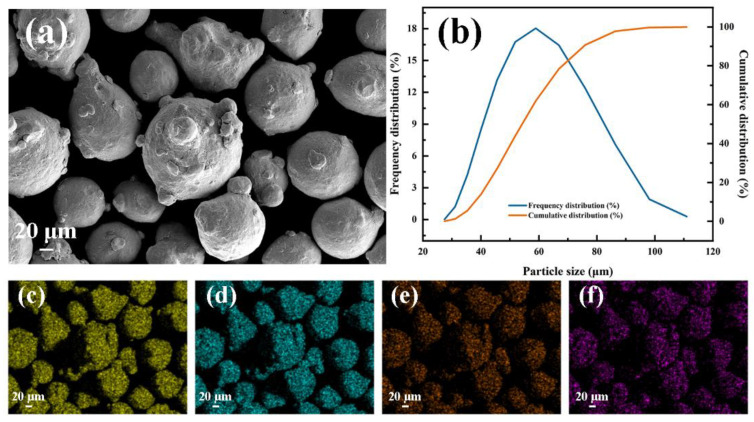
Inconel625 alloy powders: (**a**) SEM image; (**b**) particle size distribution; main element mapping: (**c**) Ni, (**d**) Cr, (**e**) Mo, (**f**) Nb.

**Figure 2 materials-15-06400-f002:**
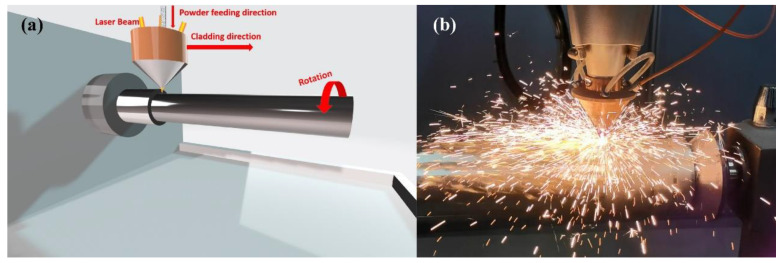
Diagram of laser cladding: (**a**) experimental platform; (**b**) the actual process.

**Figure 3 materials-15-06400-f003:**
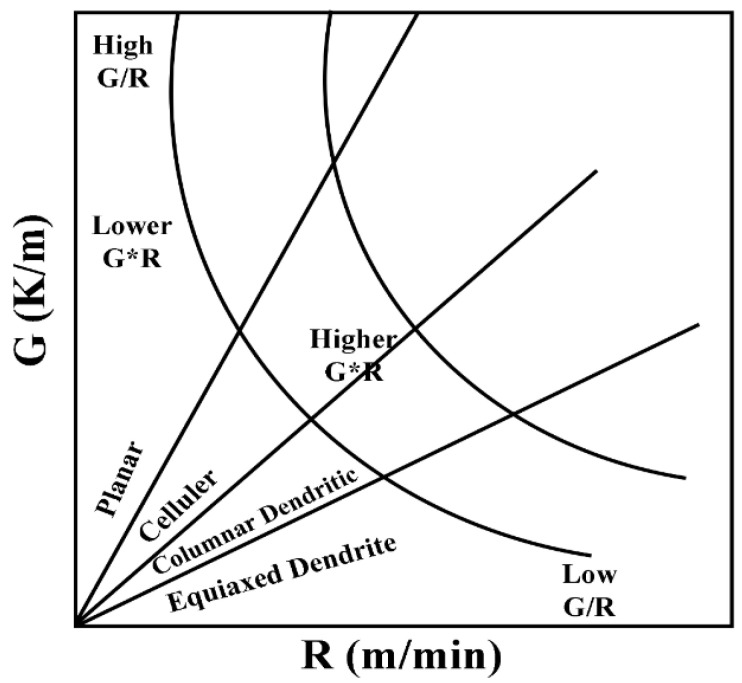
The effects of G and R on the morphology and scale of the solidified microstructure: G/R determines the morphology of the solidified structure; G*R determines the size of the solidified structure. Reprinted with permission from Ref. [31], 2022, Elsevier.

**Figure 4 materials-15-06400-f004:**
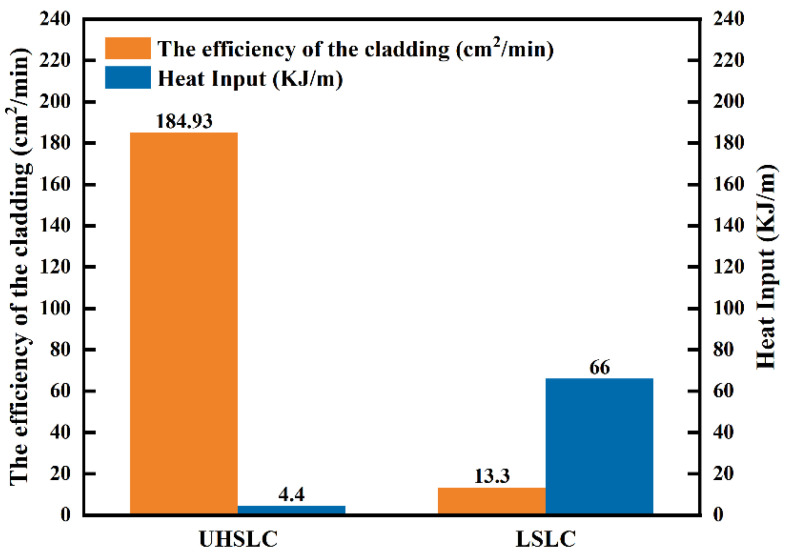
Comparison of heat input and efficiency of Inconel625 alloys coatings prepared by LSLC and UHSLC.

**Figure 5 materials-15-06400-f005:**
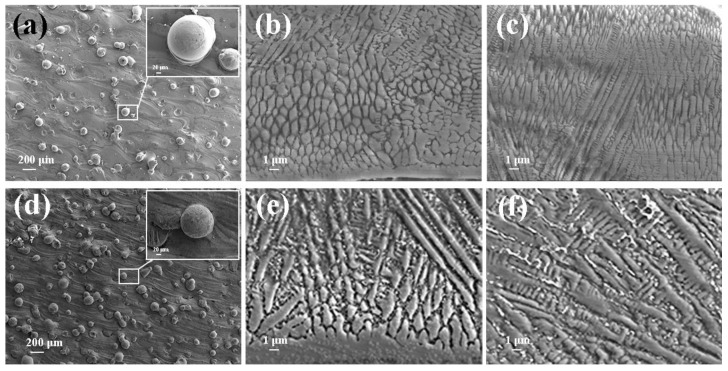
SEM image of the Inconel625 coatings prepared by UHSLC: (**a**) surface; (**b**) bottom; (**c**) top; SEM image of the Inconel625 coatings prepared by LSLC: (**d**) surface; (**e**) bottom; (**f**) top.

**Figure 6 materials-15-06400-f006:**
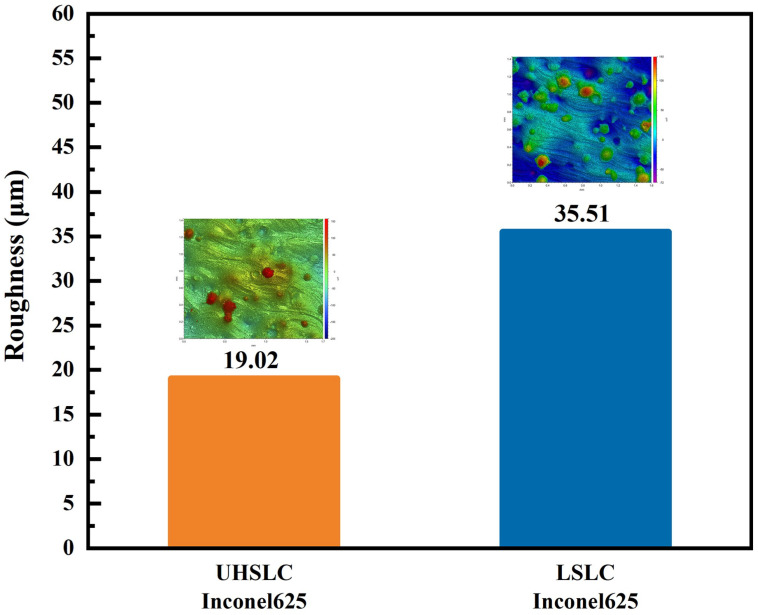
The surface roughness of Inconel625 coatings prepared by UHSLC and LSLC.

**Figure 7 materials-15-06400-f007:**
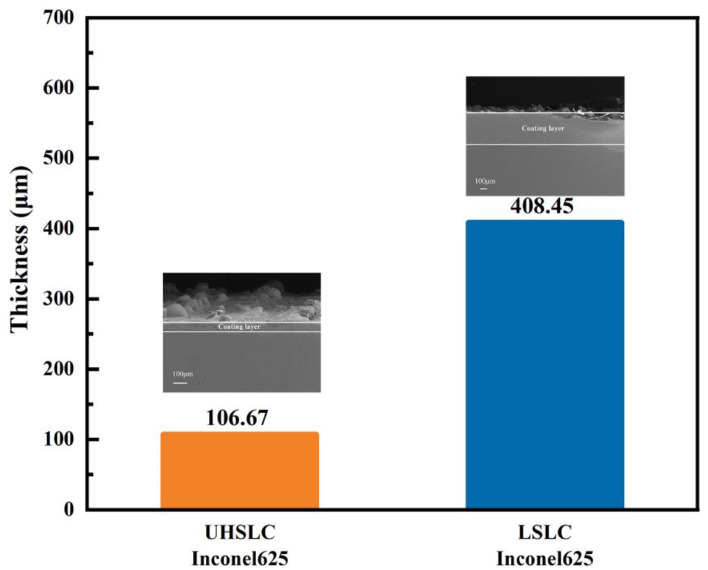
The thickness of Inconel625 alloys coatings prepared by UHSLC and LSLC.

**Figure 8 materials-15-06400-f008:**
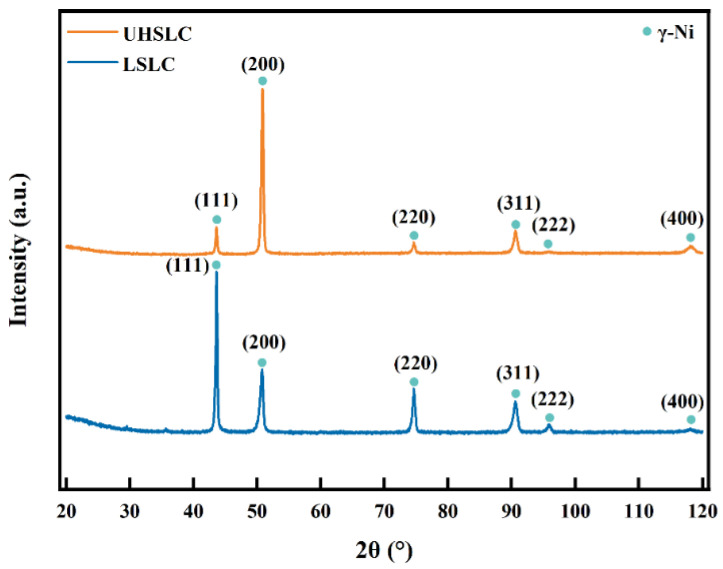
X-ray diffraction patterns of Inconel625 alloys coatings prepared by UHSLC and LSLC.

**Figure 9 materials-15-06400-f009:**
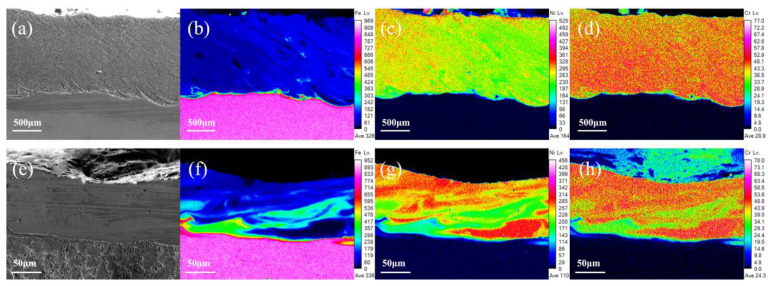
Main elements EPMA mapping image, LSLC Inconel625 alloys coatings: (**a**) SEM image; (**b**) Fe; (**c**) Ni; (**d**) Cr; UHSLC Inconel625 alloy coatings: (**e**) SEM image; (**f**) Fe; (**g**) Ni; (**h**) Cr.

**Figure 10 materials-15-06400-f010:**
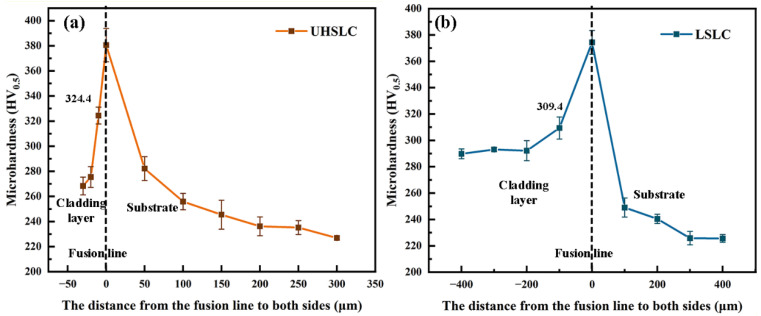
Cross-sectional microhardness distribution of Inconel625 alloys coatings: (**a**) under UHSLC; (**b**) under LSLC.

**Figure 11 materials-15-06400-f011:**
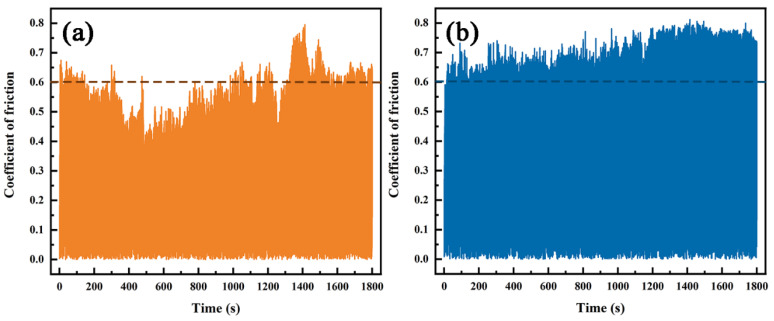
The friction coefficient curves of Inconel625 alloys coatings: (**a**) under UHSLC; (**b**) under LSLC.

**Figure 12 materials-15-06400-f012:**
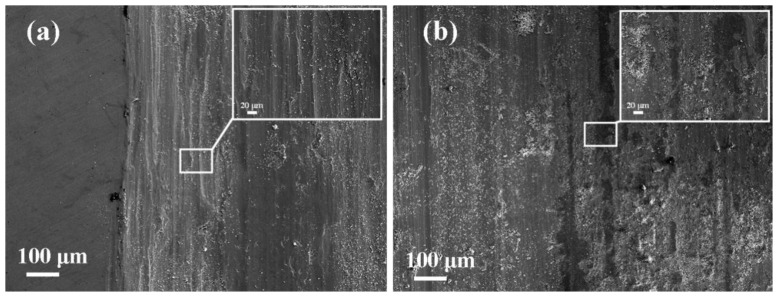
SEM images of worn surfaces of Inconel625 alloys coatings after friction and wear tests: (**a**) under UHSLC; (**b**) under LSLC.

**Figure 13 materials-15-06400-f013:**
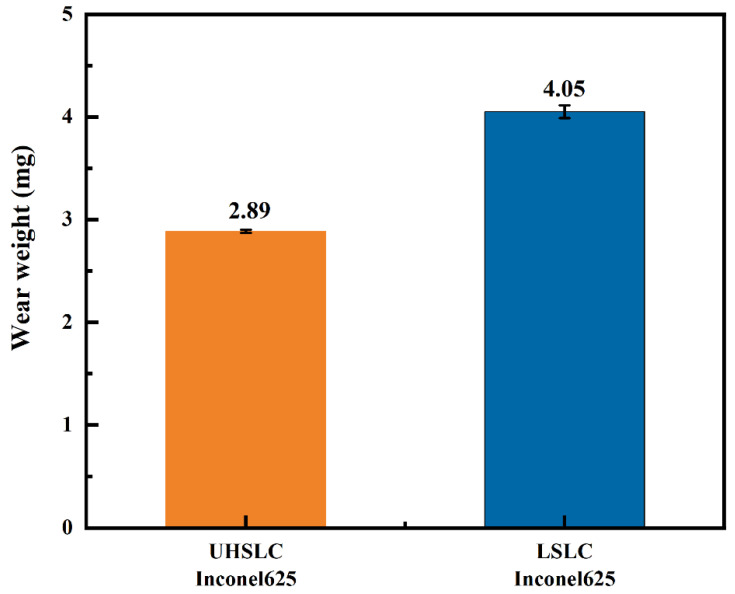
The wear mass loss of Inconel625 alloys coatings prepared by UHSLC and LSLC.

**Figure 14 materials-15-06400-f014:**
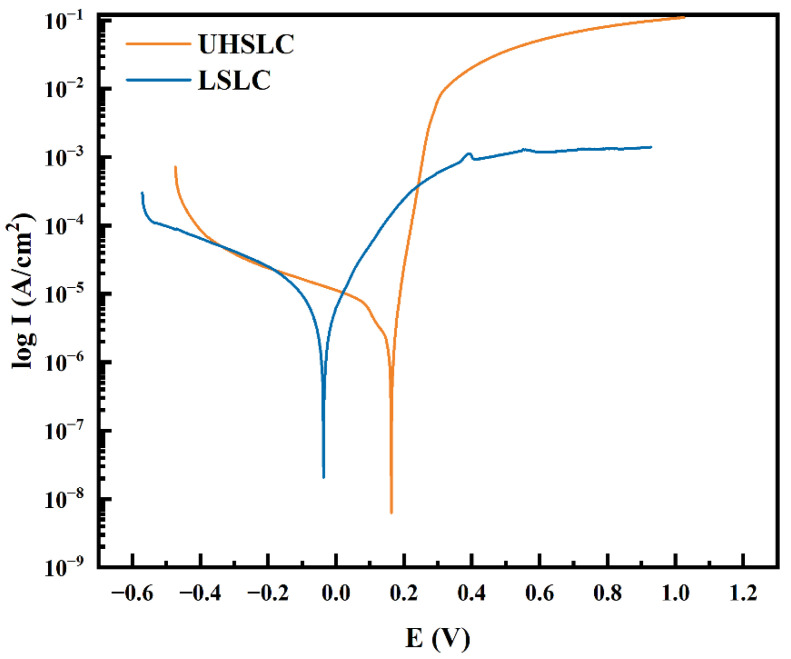
Electrochemical corrosion polarization curve of Inconel625 alloys coatings prepared by UHSLC and LSLC.

**Figure 15 materials-15-06400-f015:**
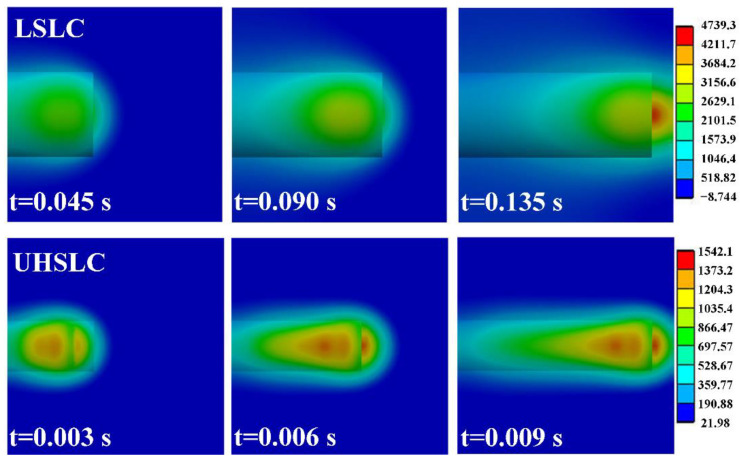
Temperature distribution at different action time LSLC: (**a**) 0.045 s; (**b**) 0.09 s; (**c**) 0.135 s; UHSLC: (**d**) 0.003 s; (**e**) 0.006 s; (**f**) 0.009 s.

**Figure 16 materials-15-06400-f016:**
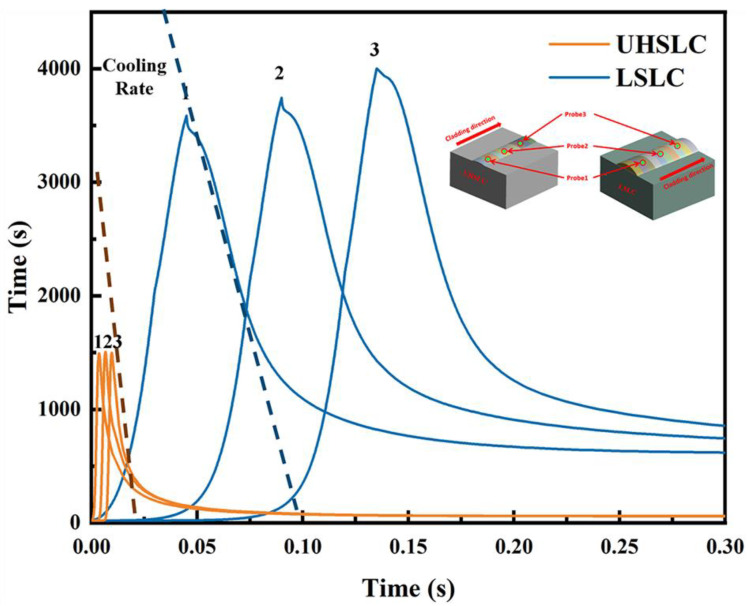
Temperature distribution of UHSLC and LSLC at different points.

**Table 1 materials-15-06400-t001:** Chemical composition (wt.%) of Inconel625 powder and 27SiMn steel substrate.

Material	Cr	C	Mn	Ni	Si	Mo	Fe	V	Co	Nb	Cu	Sd	Sn
27SiMn	<0.30	0.24–0.32	1.01	<0.30	0.96		Bal						
Inconel625	21.87		0.39	63.35		9.5	0.75		0.06	3.64	0.32	0.023	0.05

**Table 2 materials-15-06400-t002:** Processing parameters for Inconel625 alloys coatings prepared by UHSLC and LSLC.

Form	Laser Power (W)	Cladding Rate (m/min)	Powder Feeding Speed (g/min)	Overlap Rate
UHSLC LSLC	2200	30	32.3	85%
	2200	2	32.3	85%

**Table 3 materials-15-06400-t003:** Comparison of FWHM, grain size, dislocation density and micro-stress for different remelting powers.

Form	Peak Position 2θ (°)	FWHM (°)	Crystallite Size (nm)	Dislocation Density δ × 10^−3^ (nm^−2^)	a (Å)
UHSLC LSLC	50.84	0.41	21.58	2.15	3.59
	43.62	0.34	25.33	1.56	3.11

**Table 4 materials-15-06400-t004:** Polarization curve parameters of Inconel625 alloys coatings prepared by UHSLC and LSLC.

Form	Corrosion Current Density (A/cm^2^)	Self-Corrosion Potential (V)	CR (mm/year)
UHSLC	1.99 × 10^−6^	0.19	2.09 × 10^−2^
LSLC	2.64 × 10^−6^	−0.04	2.78 × 10^−2^

## Data Availability

The data that support the findings of this study are available from the corresponding author upon reasonable request.

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
