# Peer review of "A Comparative Study on Microstructure and Properties of Ultra-High-Speed Laser Cladding and Traditional Laser Cladding of Inconel625 Coatings"

_materials, 2022, doi:10.3390/ma15186400_

Round 1
Reviewer 1 Report
This manuscript reports preparation of high quality Inconel625 coating using techniques of both ultra-high-speed and traditional low-speed laser cladding. The ultra-high-speed one (UHSLC) is approved to be highly efficient and also generated Inconel625 with better qualities (crystal structure, hardness etc). The systematic examinations and analyses provide sufficient support for the conclusions of this paper. I have no further questions. So I would be happy to support the publication of this manusript in Materials.
Author Response
We are grateful for you taking the time to read our article carefully and giving us such high praise.
Reviewer 2 Report
The manuscript "A comparative study on microstructure and properties of ultrahigh-speed laser cladding and traditional laser cladding of Inconel625 coatings" has been reviewed. It deals with an experimental comparison between ultra-high speed laser cladding and traditional laser cladding on microstructure and properties.
The manuscript is almost clear, relatively new and well arranged. English acceptable.
It can be reconsidered for publication after the following major revisions:
Line 92: 46.78 (only one decimal)!
Fig. 1 b) is truncated horizontally. Please check. Futhermore is it coming from an article? Please add citation!
Line190: prepared NOT prepsred!
XRD: please specify X-Ray source, filter and wavelength.
Line 332-334: check font and size.
Fig. 15: add markers a) - f), and captions.
References (all) are not in compliance with the journal requirements.
Conclusions should be better highlighted and deriving from the results.
Author Response
Checklist and Response
Reviewer #2 (Comments to the Author):
- Line 92: 46.78 (only one decimal)!
Response: Thank you very much for your kind attention. The mentioned issue has been corrected.
- 1 b) is truncated horizontally. Please check. Futhermore is it coming from an article? Please add citation!
Response: Thank you very much for your valuable suggestions. The particle size distribution was tested by laser diffraction particle size analyzer (Mastersizer 3000, Malvern, UK), and we have redrew the Fig. 1(b).
- Line190: prepared NOT prepsred!
Response: Thank you very much for your kind attention. The mentioned issue has been corrected.
- XRD: please specify X-Ray source, filter and wavelength.
Response: Thank you very much for your valuable suggestions. The Cu Kα (λ = 1.54 Å) radiation was used in all the patterns, and graphite monochromator was used as a filter, these parameters have been added in part of 2.3 Characterization.
- Line 332-334: check font and size.
Response: Thank you very much for your kind attention. The mentioned issue has been corrected.
- 15: add markers a) - f), and captions.
Response: Thank you very much for your kind attention. The mentioned issue has been corrected.
- References (all) are not in compliance with the journal requirements.
Response: Thank you very much for your kind attention. The mentioned issue has been corrected.
- Conclusions should be better highlighted and deriving from the results.
Response: Thank you very much for your valuable suggestions. According to your directions, we have redrafted the conclusions.

Reviewer 3 Report
The current study illustrates the effect of the speed of laser cladding on the mechanical and microstructure of the Inconel625 coatings.
The mechanical and microstructure investigation is quite interesting. The major outcome results maybe have of interest to the scientific community. This study will be of high-value interest if they studied the microstructure and the mechanical structure of Inconel625 coatings by applying different speeds of laser cladding.
I recommend accepting the article after the authors maintain the following comment;
- Clarify the meaning and definition of Ultra-High speed of the laser cladding and its importance the references (11 and 12) are not enough and the authors have many arctics comparing the low speed and high speed of the laser cladding. What is the limit between the high-speed and ultra-high-speed laser cladding?
- Illustrate the parameters of the traditional laser cladding in the introduction.
- Please, illustrate the source of Inconel625 powder (is prepared in your lab? or commercial (indicate the source)?)
- Adding the errors bar in Figure 1b and table 1 with the element mapping to see the distribution of the elements before the laser cladding process.
- Add the errors bar in figure 4 and mention it in the text.
- You have to use the same magnification scale bar for comparing the morphological properties between UHSLC and LSLC (figure 5b and 5e) and figure 5c and 5f.
- Add the errors bar in figure 6 and mention it in the text.
- The XRD analysis is very poor, please include a table with (lattice parameters, crystalline phase content, and average grain size) for UHSLC and LSLC methods for better discussion you can follow the same procedures and calculations reported in (https://www.mdpi.com/2227-9040/10/6/225/htm).
- Add the errors bar in Figures 10a and 10b and mention it in the text.
- Add the errors bar in figure 13 and mention it in the text.
- Enhance the conclusion with relevant and possible applications and compare the mechanical parameters and the microstructure differences between UHSLC and LSLC.
- Minor revision for the references.
Author Response
Checklist and Response
Reviewer #2 (Comments to the Author):
- Clarify the meaning and definition of Ultra-High speed of the laser cladding and its importance the references (11 and 12) are not enough and the authors have many arctics comparing the low speed and high speed of the laser cladding. What is the limit between the high-speed and ultra-high-speed laser cladding?
Response: Thanks very much for your valuable suggestions, we have supplemented the definition of ultra-High speed laser cladding, high speed laser cladding and low speed laser cladding in the revised manuscript. According to the previous research, when the laser cladding speed reach 20-200 m/min, which is called ultra-high-speed laser cladding [1–4]. When the laser cladding speed reach 2-20 m/min, which is called high-speed laser cladding [5–7]. When the laser cladding speed is low than 2 m/min, which is called low speed laser cladding.
Whether or not the laser cladding speed reach 20 m/min is the limit between the high-speed and ultra-high-speed laser cladding.
- Illustrate the parameters of the traditional laser cladding in the introduction.
Response: Thanks very much for your valuable suggestions, we have added the relevant parameters of the traditional laser cladding in the introduction.
- Please, illustrate the source of Inconel625 powder (is prepared in your lab? or commercial (indicate the source)?)
Response: Thanks very much for your attention, the Inconel625 powder was bought from Höganäs, Sweden, relevant information has been added in line 94.
- Adding the errors bar in Figure 1b and table 1 with the element mapping to see the distribution of the elements before the laser cladding process.
Response: We appreciate your beneficial suggestions, and understand your concerns for quantitative analysis of SEM measurements. In order to reflect real particle size distribution situations of Inconel625 powder, a laser diffraction particle size analyzer (Mastersizer 3000, Malvern, UK) was used, and we have redrew the Fig. 1(b). According to your suggestion, the main element mapping has added, please see Fig. 1 (c), (d), (e) and (f).
- Add theerrors bar in figure 4 and mention it in the text.
Response: We appreciate your beneficial suggestions, however, the efficiency of the laser cladding and the heat input is a fixed value when use the same laser power, the efficiency of the cladding and the heat input, both of these were calculated by the formula below
(1)
where, η represents the cladding efficiency, A is the cladding area and T is the time taken for cladding the A area.
(2)
where, E represents the heat input, P is the laser power, and v is the heat source moving speed. So, there is no errors bar in figure 4.
- You have to use the same magnification scalebar for comparing the morphological properties between UHSLC and LSLC (figure 5b and 5e) and figure 5c and 5f.
Response: Thanks very much for your valuable suggestions, we have supplemented the same magnification scale bar SEM pictures for comparing the morphological properties between UHSLC and LSLC (figure 5b and 5e) and figure 5c and 5f.
- Add the errors bar in figure 6 and mention it in the text.
Response: Thanks very much for your valuable suggestions, we have added the errors bar in figure 6 and mention it in the text.
- The XRD analysis is very poor, please include a table with (lattice parameters, crystalline phase content, and average grain size) for UHSLC and LSLC methods for better discussion you can follow the same procedures and calculations reported in (https://www.mdpi.com/2227-9040/10/6/225/htm).
Response: Thanks very much for your valuable suggestions, a table with lattice parameters, crystalline phase content, and average grain size for XRD analysis have been added.
- Add the errors bar in Figures 10a and 10b and mention it in the text.
Response: Thanks very much for your valuable suggestions, we have added the errors bar in Figures 10a and 10b and mention it in the text.
- Add the errors bar in figure 13 and mention it in the text.
Response: Thanks very much for your valuable suggestions, we have added the errors bar in Figure 13 and mention it in the text.
- Enhance the conclusion with relevant and possible applications and compare the mechanical parameters and the microstructure differences between UHSLC and LSLC.
Response: Thanks very much for your valuable suggestions. According to your directions, we have redrafted the conclusions.
- Minor revision for the references.
Response: Thanks very much for your valuable suggestions, the references have been revised.
References
- Li, L.; Shen, F.; Zhou, Y.; Tao, W. Comparative Study of Stainless Steel AISI 431 Coatings Prepared by Extreme-High-Speed and Conventional Laser Cladding. Journal of Laser Applications2019, 31, 042009, doi:10.2351/1.5094378.
- Schopphoven, T.; Gasser, A.; Wissenbach, K.; Poprawe, R. Investigations on Ultra-High-Speed Laser Material Deposition as Alternative for Hard Chrome Plating and Thermal Spraying. Journal of Laser Applications2016, 28, 022501, doi:10.2351/1.4943910.
- Schopphoven, T.; Gasser, A.; Backes, G. EHLA: Extreme High-Speed Laser Material Deposition. Laser Technik Journal2017, 14, 45–45, doi:10.1002/latj.201770308.
- Liu, M.; Jiang, H.; Chang, G.; Xu, Y.; Ma, F.; Xu, K. Effect of Laser Remelting on Corrosion and Wear Resistance of Fe82Cr16SiB Alloy Coatings Fabricated by Extreme High-Speed Laser Cladding. Materials Letters2022, 325, 132823, doi:10.1016/j.matlet.2022.132823.
- Hu, Z.; Li, Y.; Lu, B.; Tan, N.; Cai, L.; Yong, Q. Effect of WC Content on Microstructure and Properties of High-Speed Laser Cladding Ni-Based Coating. Optics & Laser Technology2022, 155, 108449, doi:10.1016/j.optlastec.2022.108449.
- Liu, J.; Li, Y.; He, P.; Tan, N.; Li, Q.; Zhang, G.; Lu, B. Microstructure and Properties of ZrB2-SiC Continuous Gradient Coating Prepared by High Speed Laser Cladding. Tribology International2022, 173, 107645, doi:10.1016/j.triboint.2022.107645.
- Chong, Z.; Sun, Y.; Cheng, W.; Huang, L.; Han, C.; Ma, X.; Meng, A. Laser Remelting Induces Grain Refinement and Properties Enhancement in High-Speed Laser Cladding AlCoCrFeNi High-Entropy Alloy Coatings. Intermetallics2022, 150, 107686, doi:10.1016/j.intermet.2022.107686.
- Yuan, W.; Li, R.; Chen, Z.; Gu, J.; Tian, Y. A Comparative Study on Microstructure and Properties of Traditional Laser Cladding and High-Speed Laser Cladding of Ni45 Alloy Coatings. Surface and Coatings Technology2021, 405, 126582, doi:10.1016/j.surfcoat.2020.126582.

Round 2
Reviewer 2 Report
The improved manuscript can be accepted in the present form.
Reviewer 3 Report
The authors complete all my recommended comments and the manuscript can be published in its present form.